# Systematic Assessment of Phonon and Optical Characteristics for Gas-Source Molecular Beam Epitaxy-Grown InP$_{1-x}$Sb$_x$/n-InAs Epifilms

**Devki N. Talwar** [1,2,*] and **Hao-Hsiung Lin** [3]

1. Department of Physics, University of North Florida, 1 UNF Drive, Jacksonville, FL 32224, USA
2. Department of Physics, Indiana University of Pennsylvania, 975 Oakland Avenue, 56 Weyandt Hall, Indiana, PA 15705, USA
3. Graduate Institute of Electronics Engineering, Department of Electrical Engineering, National Taiwan University, Taipei 10617, Taiwan; hhlin@ntu.tw.edu
* Correspondence: d.talwar@unf.edu; Tel.: +1-724-762-7719

**Abstract:** Experimental and theoretical assessments of phonon and optical characteristics are methodically accomplished for comprehending the vibrational, structural, and electronic behavior of InP$_{1-x}$Sb$_x$/n-InAs samples grown by Gas-Source Molecular Beam Epitaxy. While the polarization-dependent Raman scattering measurements revealed InP-like doublet covering optical modes ($\omega_{LO}^{InP} \sim 350$ cm$^{-1}$, $\omega_{TO}^{InP} \sim 304$ cm$^{-1}$) and phonons activated by disorders and impurities, a single unresolved InSb-like broadband is detected near ~195 cm$^{-1}$. In InP$_{1-x}$Sb$_x$, although no local vibrational (In*Sb*:P; x → 1) and gap modes (In*P*:Sb; x → 0) are observed, the Raman line shapes exhibited large separation between the optical phonons of its binary counterparts, showing features similar to the phonon density of states, confirming "two-mode-behavior". Despite the earlier suggestions of large miscibility gaps in InP$_{1-x}$Sb$_x$ epilayers for x between 0.02 and 0.97, our photoluminescence (PL) results of energy gaps insinuated achieving high-quality single-phase epilayers with x~0.3 in the miscibility gap. Complete sets of model dielectric functions (MDFs) are obtained for simulating the optical constants of binary InP, InSb, and ternary InP$_{1-x}$Sb$_x$ alloys in the photon energy ($0 \leq E \leq 6$ eV) region. Detailed MDF analyses of refractive indices, extinction coefficients, absorption and reflectance spectra have exhibited results in good agreement with the spectroscopic ellipsometry data. For InP$_{0.67}$Sb$_{0.33}$ alloy, our calculated lowest energy bandgap E$_0$ ~ 0.46 eV has validated the existing first-principles calculation and PL data. We feel that our results on Raman scattering, PL measurements, and simulations of optical constants provide valuable information for the vibrational and optical traits of InP$_{1-x}$Sb$_x$/n-InAs epilayers and can be extended to many other technologically important materials.

**Keywords:** gas-source molecular beam epitaxy; Raman scattering spectroscopy; photoluminescence; model dielectric functions; optical constants; two-phonon-mode behavior



## 1. Introduction

Antimony (Sb)-based high-electron, high–hole mobility III–V binary (InSb, GaSb, AlSb), ternary (InAlSb, InGaSb, GaAlSb), and quaternary (InGaAlSb, InGaPSb, InAlPSb, GaAlPSb, InGaPSb) alloys and heterostructures (quantum wells (QWs) including superlattices (SLs)) have attracted a great deal of attention due to their promises for developing next-generation high-speed infrared opto-electronic and thermo-electric devices [1–15]. Among others, the indium-based pnictides (InX; X = P, As, and Sb) have recently gained considerable importance. Although the lowest bandgap E$_g$ ($\equiv$0.18 eV) of InSb material is critical for mid-infrared optoelectronics, ultrathin films, QWs, and SLs involving InP (E$_g$ $\equiv$ 1.35 eV) and InAs (E$_g$ $\equiv$ 0.35 eV) are being used to design metal oxide–semiconductor field-effect transistors (MOSFETs), bipolar junction transistors (BJTs), multi-junction solar cells, light-emitting/

receiving devices, infrared detectors (IDs), Shockley diodes (SDs), long-wavelength laser diodes (LDs), photodetectors (PDs), thermoelectric generators (TEGs), etc. [1–16]. According to Talazac et al. [17], several Schottky devices have been integrated recently into different ecosystems for detecting radiation, harmful gases, and pollutants (e.g., ozone $O_3$, nitrogen oxide $NO_3$, etc.) and monitoring chemical exposure in the environment. Other devices are also used in satellite communication equipment, high-speed/low-power circuits for bio-medical diagnostics, drug analysis, etc.

As compared to the conventional III–V semiconductors, the growth of InP(As)Sb and/or In(Ga,Al)PSb materials has been and still is quite difficult [18]. As active cladding layers, the exploitation of ultrathin $InP_{1-x}Sb_x$ alloy films is considered indispensable for mid-infrared (MIR) laser sources. Due to a large ~10.4% lattice mismatch between InP and InSb, the growth of defect-free crystalline thin films of $InP_{1-x}Sb_x$ ternary alloys is quite challenging. Other issues of preparing Sb-based device structures are linked to the (i) low vapor pressure of Sb, (ii) limitations of the kinetically controlled growth regime, (iii) inexistence of chemically stable hydrides as precursors, and (iv) lack of insulating substrates [16–38]. Despite these difficulties, several attempts have been made to prepare Sb-based alloys, QWs, and SLs by using liquid-phase epitaxy (LPE), organo-metallic vapor-phase epitaxy (OMVPE), and molecular beam epitaxy (MBE) techniques [39–41]. There are still a few intrinsic issues that inhibit the design of several important device structures. However, solutions to these problems can be achieved by exploiting appropriate experimental [1–17] and/or theoretical techniques [18].

Despite many efforts made to grow diverse Sb-based structures, there are limited studies assessing their optical and vibrational characteristics. The experimental methods used for material characterizations include high-resolution X-ray diffraction (HR-XRD) [39,40], Fourier transform infrared (FTIR) [42–44], Raman scattering spectroscopy (RSS) [45,46], photoluminescence (PL) [25–27,47], inelastic neutron scattering (INS) [48,49], inelastic X-ray scattering (IXS) [50], reflection high-energy electron diffraction (RHEED), transmission electron microscopy (TEM) [35–38], spectroscopic ellipsometry (SE) [28,29], etc. While HR-XRD [39,40], FTIR [42–44], and RSS have played [45,46] vital roles in appraising the alloy composition x, film thickness d, interfacial structure, and surface relaxation of atoms, their exploitation to assess basic properties of Sb-based ternary and quaternary alloys remained, however, surprisingly enigmatic.

For $InP_{1-x}Sb_x$ alloys, there exist few experimental and theoretical studies, especially on the physics of those attributes which ascertain their prominence at a practical level [1–17]. In the far-infrared (FIR) region, 5 meV $\leq$ E $\leq$ 100 meV, although SE is recognized as an efficient method for exploring lattice dynamics and free carrier concentration of binary materials, the method has not yet been applied to ternary alloys. Earlier analysis of RSS data on vibrational properties of $InP_{1-x}Sb_x$ using a modified random-element iso-displacement (MREI) model envisioned a two-phonon-mode behavior [25–27]. However, no gap (In*P*:Sb for x $\to$ 0) or localized vibrational mode (LVM) (In*Sb*:P for x $\to$ 1) was detected near the limiting values of x. Extensive PL measurements and extended X-ray absorption fine-structure (EXAFS) results on Gas-Source Molecular Beam Epitaxy (GS-MBE)-grown $InP_{1-x}Sb_x$/n-InAs samples have recently provided valuable electronic and structural characteristics [51]. On the contrary, no attempts have been made to comprehend the optical properties of $InP_{1-x}Sb_x$ alloys in the near-IR (NIR) to ultraviolet (UV) energy range.

The purpose of this paper is to report the results of both experimental (Section 2) and theoretical (Section 3) studies to assess the phonon and optical traits of nearly ~1 μm thick $InP_{1-x}Sb_x$/n-InAs samples grown by the GS-MBE technique (Section 2.1). The TEM method was employed earlier for investigating the structural and chemical distributions of atoms. The composition of Sb in $InP_{1-x}Sb_x$ alloys has also been achieved by using electron probe microanalysis (EPMA). Room-temperature (RT) Hall measurements with van der Pauw methods have been performed to assess the electrical properties of Be- and Si-doped $InP_{1-x}Sb_x$ epilayers [39,40]. Here, we employed a Reinshaw InVia Raman (Section 2.2) spectrometer with a diode-pumped solid-state (DPSS) 532 nm laser as an

excitation source for measuring the composition-dependent longitudinal optical ($\omega_{LO}$) and transverse optical ($\omega_{TO}$) phonons in $InP_{1-x}Sb_x$/n-InAs samples. The PL measurements are also reported by calibrating the radiation lines of a Xe lamp and employing the SPEX 500M monochromator (Section 2.3). Like HR-XRD, the PL studies on $InP_{1-x}Sb_x$/n-InAs samples with smaller values of x ($\equiv$0.10, 0.16, 0.17) have confirmed the two energy phases, and for x $\geq$ 0.30, it provided a single energy phase. From a theoretical standpoint (Section 3.1), a classical "Drude-Lorentz" model is adopted by incorporating optical $\omega_{LO}$, $\omega_{TO}$ phonons to simulate (Section 3.2) the complex dielectric functions $\left[ \widetilde{\epsilon}(\omega), \text{or } \widetilde{n}(\omega) \right]$ in the FIR $\rightarrow$ MIR (5 meV $\leq$ E $\leq$ 100 meV) energy region for both binary and ternary alloys. Although the optical measurements by SE in the NIR $\rightarrow$ UV (0. 5 eV to 6.0 eV) energy regions are known for InP and InSb materials [22,23], little to no information is available for the ternary $InP_{1-x}Sb_x$ alloys. From technical and scientific perspectives, it is crucial to derive analytical expressions for simulating the optical constants of both binary and ternary alloys. The systematic assessment of refractive indices $\widetilde{n}(E)$ and absorption coefficients $\alpha(E)$ is especially important for selecting apposite materials in structural designs and the optimization of different optoelectronic devices. By adopting Adachi's [37,38] optical dispersion mechanisms and using a modified dielectric function (MDF) approach (Sections 3.3 and 3.3.1, Sections 3.3.2 and 3.3.3), we have systematically analyzed the complex dielectric functions of direct bandgap binary InP, InSb materials in the NIR $\rightarrow$ UV region. This methodology is meticulously extended to obtain the optical constants of $InP_{1-x}Sb_x$ ternary alloys for any arbitrary composition x and photon energy E [$\equiv$ $\hbar\omega$]. For $InP_{0.67}Sb_{0.33}$ alloy, our MDF simulation has provided the lowest bandgap energy $E_0$(near ~0.46 eV), in reasonably good agreement with the existing first-principles calculations [18] and PL measurements. Theoretical results of phonons and optical traits are compared/contrasted and discussed against the existing experimental data (Sections 4.1, 4.1.1, 4.1.2, 4.2, 4.2.1 and 4.2.2), with concluding remarks presented in Section 5. We strongly feel that our RSS and PL studies have provided valuable information on the phonon and optical traits of $InP_{1-x}Sb_x$/n-InAs (001) epilayers and can be extended to many other technologically important materials.

## 2. Experimental Procedures

Different $InP_{1-x}Sb_x$ epilayers of thickness d ($\equiv$~1 μm) with diverse Sb (see Table 1) compositions x are prepared on n-InAs substrate by using GS-MBE technique (Section 2.1). Hall effect and van der Pauw measurements were reported earlier on several Be- and Si-doped $InP_{1-x}Sb_x$ samples to assess their electrical properties [39,40]. Several experimental characterization tools are exploited (viz., RSS and PL) here to investigate the fundamental features of $InP_{1-x}Sb_x$/n-InAs epilayers (Sections 2.2 and 2.3), including the phonon and optical traits. Careful analyses of RSS, PL, and SE results are performed (Section 3) to attain valuable characteristics for the binary InP, InSb and ternary $InP_{1-x}Sb_x$ alloys.

**Table 1.** Different high-quality, undoped samples of nearly one-micron-thick $InP_{1-x}Sb_x$ on n-InAs (001) substrate (see text) grown using the gas-source molecular beam epitaxy (GS-MBE) technique (see text).

| Sample | x [a] | 1 − x |
|---|---|---|
| # T0 | 0.10 | 0.90 |
| # T1 | 0.16 | 0.84 |
| # T2 | 0.17 | 0.83 |
| # T3 | 0.30 | 0.70 |
| # T4 | 0.33 | 0.67 |
| # T5 | 0.36 | 0.64 |
| # T6 | 0.37 | 0.63 |
| # T7 | 0.48 | 0.52 |

[a] Refs. [39,40].

*2.1. GS-MBE Growth of InP$_{1-x}$Sb$_x$/n-InAs (001) Epifilms*

Several high-quality InP$_{1-x}$Sb$_x$ ($0.1 \leq x \leq 0.48$) samples were grown on n-InAs (001) substrates. A gas-source VG-V80H MBE system was employed by setting the substrate temperature between ~470 °C and 480 °C. Pure PH$_3$ gas was injected into a gas cell, in which phosphine was cracked at 1000 °C to deliver P$_2$. An EPI Sb cell was used as the Sb source and its cracking zone was set at ~1050 °C to supply a mixed beam of Sb and Sb$_2$. A conventional thermal effusion K-cell was used to provide the flux flow of group III In. All the InP$_{1-x}$Sb$_x$/n-InAs epifilms were prepared at a growth rate of ~1 μm/h. The in situ reflection high-energy electron diffraction (RHEED) technique was employed earlier to monitor the surface reconstructions [39,40].

*2.2. Raman Scattering*

Raman scattering spectroscopy measurements are performed on several GS-MBE-grown InP$_{1-x}$Sb$_x$/n-InAs samples by exploiting a Reinshaw InVia spectrometer with a 532 nm DP-SSL as an excitation source. On each sample, the incident power is set at 60 mW after allowing the optical losses. A holographic notch filter is employed to block unwanted reflections. The results of composition-dependent optical phonons are obtained for InP$_{1-x}$Sb$_x$ alloys in the frequency range of ~40 cm$^{-1}$ to 800 cm$^{-1}$. Several scans are considered on each sample at different locations with a run of 16 accumulations for reducing the effects of noise. Despite setting the exposure time of 10 s to minimize heating effects on each sample, it remained inevitable that the local heating would cause broadening in the observed phonon bands (Section 4.1).

Polarization-dependent Raman spectra have also been recorded (Section 4.1.1), keeping the incident beam fixed. We analyzed the scattered beam by a Polaroid sheet combined with a half-wave plate placed in front of the entrance slit for recording the spectra with two different polarization geometries, $z(Y, X)\bar{z}$ and $z(X, X)\bar{z}$, where X, Y, and Z denote the coordinates of (100), (010), and (001) directions with respect to the n-InAs substrate. On InP$_{1-x}$Sb$_x$/n-InAs samples, the results with spectral linewidth ~2.0 cm$^{-1}$ are attributed to the scattering from optical phonons activated by disorders and/or impurities. Clearly, the observed Raman intensity line shapes are represented (Sections 4.1 and 4.1.1) as the phonon density of states, corroborating the two-phonon mode behavior in InP$_{1-x}$Sb$_x$ alloys.

*2.3. Photoluminescence*

PL studies have offered a simple, non-destructive method for assessing the electronic bandgaps (E$_g^{PL}$) of binary and ternary alloy semiconductors. On GS-MBE-grown InP$_{1-x}$Sb$_x$/n-InAs samples, we perform PL measurements in a wide range of temperatures (10 K–300 K) by calibrating the radiation lines of a Xe lamp and exploiting the SPEX 500M monochromator (Section 4.2). A 532 nm DP-SSL excitation source is used by adjusting its maximum output power at 110 mW. The luminescence dispersed from each sample by a monochromator is collected by a liquid-nitrogen-cooled InSb detector. A correct beam path is adapted by using two pin holes and placing a lens to focus the incoming laser beam at a spot of nearly ~0.1 mm diameter. For each epifilm, the results of PL spectra at 13 K are recorded by exploiting the standard lock-in techniques. CaF$_2$ lenses and optical windows are used for preventing absorption in the spectral range of water vapor (~2.7-μm) and carbon monoxide (~4.1-μm). Like HR-XRD, our PL measurements on InP$_{1-x}$Sb$_x$/n-InAs samples have confirmed two energy phases with smaller values of x ($\equiv$0.10, 0.16, 0.17), while for x $\geq$ 0.30, the study provided a single energy phase. Our PL measurements for a sample with x = 0.33 offered the lowest bandgap energy E$_0$ (near ~0.49 eV), in very good agreement with the MDF calculations (Section 4.2).

## 3. Theoretical Background

As compared to the reflected intensity measurements, the SE is a model-based approach where one compares the experimental data by using accurate and methodical calculations (Sections 3.1–3.3). The SE parameters ($\Psi$, $\Delta$) are generally related to the ratio ρ

of complex Fresnel reflection coefficients $\widetilde{r}^p$ and $\widetilde{r}^s$ of incident light polarized parallel ( | | ) and perpendicular ($\perp$) to the plane of incidence [22,23]:

$$\rho \equiv \frac{\widetilde{r}^p}{\widetilde{r}^s} = \tan \Psi \exp(i\Delta). \tag{1}$$

Measurements of film thickness by in-line SE methodology have an important role in monitoring the epitaxial growth process. To extract the optical constants and thickness of the layer structured materials, one needs to establish reasonably accurate models for simulating the complex dielectric functions $\widetilde{\varepsilon}(\omega)$ [refractive indices $\widetilde{n}(\omega)$] for both the epifilms and substrates in the appropriate FIR $\rightarrow$ MIR $\rightarrow$ UV energy E [$\equiv \hbar\omega$] regions.

*3.1. Optical Constants*

Optical properties [viz., refractive index $n(\omega)$, extinction coefficient $\kappa(\omega)$, reflectivity $R(\omega)$, absorption coefficient $\alpha(\omega)$, etc.] of polar semiconductors are strongly associated with the complex dielectric function $\widetilde{\varepsilon}(\omega)$ [or $\widetilde{n}(\omega)$]. Numerically, one can obtain $\widetilde{n}(\omega)$ by using $\widetilde{\varepsilon}(\omega)$ and expressing it as follows: [38]

$$\widetilde{n}(\omega) = n(\omega) + i\kappa(\omega) = \sqrt{\widetilde{\varepsilon}(\omega)}, \tag{2}$$

where the two constants $n(\omega)$ and $\kappa(\omega)$ have real, positive numbers and are usually estimated from different experimental measurements such as SE, IR, etc. [22,23,38]. The values of these constants can be evaluated by exploiting: [38]

$$n(\omega) = \left[ \frac{\left(\varepsilon_1^2 + \varepsilon_2^2\right)^{1/2} + \varepsilon_1}{2} \right]^{1/2}, \tag{3a}$$

$$\kappa(\omega) = \left[ \frac{\left(\varepsilon_1^2 + \varepsilon_2^2\right)^{1/2} - \varepsilon_1}{2} \right]^{1/2}. \tag{3b}$$

Again, by incorporating them, it is possible to evaluate the reflectivity $R(\omega)$ and absorption coefficient $\alpha(\omega)$ : [38]

$$R(\omega) = \frac{\left[ (n(\omega) - 1)^2 + \kappa(\omega)^2 \right]}{\left[ (n(\omega) + 1)^2 + \kappa(\omega)^2 \right]}, \tag{4a}$$

$$\alpha(\omega) = \frac{4\pi}{\lambda}\kappa(\omega), \tag{4b}$$

where $\lambda$ is the wavelength of light in vacuum. The real $\varepsilon_1(\omega)$ and imaginary $\varepsilon_2(\omega)$, parts of the dielectric function $\widetilde{\varepsilon}(\omega)$ of materials are also associated with the Kramers–Krönig relations via the critical point (CP) energies of their electronic energy-band structures $E_j\left(\vec{q}\right)$ : [38]

$$\varepsilon_1(\omega) - 1 = \frac{2}{\pi}\int_0^\infty \frac{\omega'\varepsilon_2(\omega')}{(\omega')^2 - \omega^2}d\omega', \tag{5a}$$

$$\varepsilon_2(\omega) = -\frac{2\omega}{\pi}\int_0^\infty \frac{\varepsilon_1(\omega')}{(\omega')^2 - \omega^2}d\omega'. \tag{5b}$$

One must note that the optical joint density of states for the semiconductor materials becomes large in the vicinity of CP energy transitions in the Brillouin zone (BZ) [28].

*3.2. Dielectric Properties in the FIR → NIR Region*

For $InP_{1-x}Sb_x$ ternary alloys, the contribution of lattice phonons to the dielectric response function $\widetilde{\varepsilon}_L(\omega)$ in the FIR → NIR region can be evaluated by exploiting a classical "Drude-Lorentz" model [38]:

$$\widetilde{\varepsilon}_L(\omega) = \varepsilon_{\infty x}\left[1 - \frac{\omega_P^2}{\omega(\omega + i\gamma_P)}\right] + \sum_{j=1,2}\frac{S_{jx}\omega_{TOxj}^2}{\omega_{TOjx}^2 - \omega^2 - i\Gamma_{j_x}\omega}. \tag{6}$$

In Equation (6), the term $\varepsilon_{\infty x}$ represents a weighted high-frequency dielectric function; $\omega_{TOjx}$ signify the InSb-like and InP-like TO-mode frequencies; $S_{jx}$ represents the oscillator strengths; $\gamma_{jx}$ represents the broadening values of TO phonons; $\omega_P$ implies the plasma frequency and $\gamma_P$ is its damping constant. The plasma frequency $\omega_P\left(\equiv \sqrt{\frac{4\pi\eta e^2}{m^*\varepsilon_\infty}}\right)$ and $\gamma_P\left(\equiv \frac{e}{m^*\mu}\right)$ of free carriers (electrons or holes) in doped materials can be assessed from the effective mass $m^*$, the carrier concentration $\eta$, the magnitude of the electron charge e, and mobility $\mu$. To simulate $n(\omega), \kappa(\omega), \varepsilon_1(\omega), \varepsilon_2(\omega), R(\omega)$, and $\alpha(\omega)$ for $InP_{1-x}Sb_x/n\text{-}InAs$ epilayers, the required $\widetilde{\varepsilon}_L(\omega)$ of the substrate is calculated independently. To attain the best-fit parameter values in Equation (6), we followed an efficient Levenberg–Marquardt [52] algorithm and used the non-linear simulations to minimize the error function $\Xi$ over *n* data points:

$$\Xi = \frac{1}{n}\sum_i^n \left|\mathfrak{R}_i^{exp} - \mathfrak{R}_i^{cal}\right|^2, \tag{7}$$

where $\mathfrak{R}_i^{exp}$ and $\mathfrak{R}_i^{cal}$ are the experimental and calculated values, respectively.

*3.3. Dielectric Properties in the NIR → UV Region*

The indium pnictides (InP, InAs, and InSb) are direct-bandgap semiconductors and crystallize in the zb $(T_d^2)$ crystal structure with $F\bar{4}3m$ space group. In Figure 1, we reproduced the electronic energy-band structure $E_j\left(\vec{q}\right)$ of InAs by Kim et al. [28] along the high-symmetry directions in the BZ using a full-potential linearized augmented Slater-type orbital (LASTO) method. The locations of inter-band transitions labeled by vertical arrows in Figure 1 for InAs are equally applicable to InP and InSb materials, as well. These CP energy transitions have played important roles in the analysis of the optical spectra.

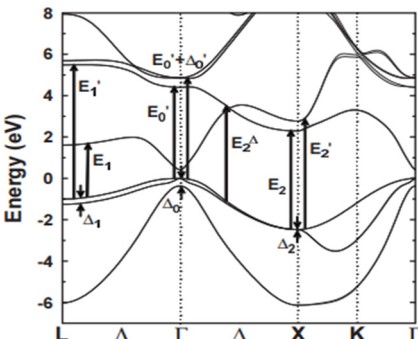

**Figure 1.** Electronic energy-band structure of direct-bandgap InAs (Ref. [28]) along high-symmetry directions in the Brillouin zone. The calculations are performed using a full-potential linearized augmented Slater-type orbital (LASTO) method (see text).

Following Adachi [37,38], we calculated the dielectric functions $\widetilde{\varepsilon}(E)$ of the binary InP and InSb materials by using the sum of different terms, each of which has an explicit function of the critical-point energies (Sections 3.3.1–3.3.3). In the NIR → UV region, the CPs that provided major contributions in the simulations of continuous model dielectric functions for InP and InSb are linked to the inter-band transitions $E_0, E_0 + \Delta_0, E_1,$

$E_1 + \Delta_1, E_0', E_0' + \Delta_0', E_2$ and $E_2'$ (Figure 1) of their band structures $E_j\left(\vec{q}\right)$ [28]. In the following subsections, we have briefly summarized the appropriate expressions used for evaluating $\widetilde{\varepsilon}(E)$, which required three fitting parameters: (a) the energy, (b) strength, and (c) broadening parameters per CP of InP and InSb [37,38].

### 3.3.1. $E_0$ and $E_0 + \Delta_0$ Transition

The $E_0$ and $E_0 + \Delta_0$ transitions are three-dimensional (3D) $M_0$ CPs and occur in InP (for instance) at photon energies ~1.35 eV and ~1.45 eV at RT (see Table 2). Assuming the bands to be parabolic, the contribution of these energy gaps to $\widetilde{\varepsilon}(E)$ can be calculated by using [37,38]

$$\widetilde{\varepsilon}^{\,a}(E) = A_0^{-1.5}\left\{ f(\chi_0) + \frac{1}{2}\left[ \frac{E_0}{E_0 + \Delta_0} \right]^{1.5} f(\chi_{s.o.}) \right\}, \tag{8}$$

with

$$f(\chi_0) = \chi_0^{-2}\left[ 2 - (1 + \chi_0)^{0.5} - (1 - \chi_0)^{0.5} \right], \tag{9a}$$

$$f(\chi_{s.o.}) = \chi_{s.o.}^{-2}\left[ 2 - (1 + \chi_{s.o.})^{0.5} - (1 - \chi_{s.o.})^{0.5} \right], \tag{9b}$$

$$\chi_0 = (\hbar\omega + i\Gamma)/E_0, \tag{9c}$$

$$\chi_{s.o.} = \frac{\hbar\omega + i\Gamma}{(E_0 + \Delta_0)}. \tag{9d}$$

**Table 2.** The best-fitted MDF parameters are exploited in the calculations of optical constants for binary InSb and InP materials (see text).

| MDF Parameters | InSb | InP |
|---|---|---|
| $E_0$ (eV) | 0.18 | 1.35 |
| $E_0 + \Delta_0$ (eV) | 0.99 | 1.45 |
| $\Gamma_0$ (eV) | 0.015 | 0.23 |
| $A_0$ (eV)$^{1.5}$ | 0.19 | 7.1 |
| $A_{0x}$ (eV$^{-1}$) | 0.007 | 0.009 |
| $E_1$ (eV) | 1.86 | 3.1 |
| $E_1 + \Delta_1$ (eV) | 2.37 | 3.31 |
| $\Gamma_1$ (eV) | 0.19 | 0.34 |
| $B_1$ | 4.10 | 2.94 |
| $B_2$ | 2.70 | 0.96 |
| $B_{1x}^1$ (eV) | 1.87 | 1.64 |
| $B_{1sx}^1$ (eV) | 0.98 | 0.82 |
| $E_2$ (eV) | 3.85 | 4.70 |
| $E_0'$ (eV) | 3.26 | 4.2 |
| $E_0' + \Delta_0'$ (eV) | 3.63 | 4.40 |
| $E_1'$ (eV) | 5.11 | 5.5 |
| $C_2$ | 2.35 | 0.22 |
| $C_0'$ | 0.50 | 0.50 |
| $C_{\Delta 0}'$ | 0.35 | 0.70 |
| $C_1'$ | 0.34 | 0.36 |
| $\Gamma_2$ (eV) | 0.17 | 0.12 |
| $\Gamma_0'$ (eV) | 0.18 | 0.18 |
| $\Gamma_{\Delta 0}'$ (eV) | 0.18 | 0.18 |
| $\Gamma_1'$ (eV) | 0.18 | 0.18 |
| $\varepsilon_{1\infty}$ | 1.20 | 0.50 |

In Equations (8) and (9a)–(9d), the terms A and $\Gamma$ are the strength and broadening parameters, respectively, for the $E_0$ and $E_0 + \Delta_0$ energy transitions. The discrete series

(n = 1, 2, 3, . . .) of exciton lines at $E_0$ and $E_0 + \Delta_0$ gaps can be written with the Lorentzian line shape by using [38]

$$\tilde{\varepsilon}^{a1}(E) = \sum_{n=1}^{\infty} \frac{A_{0x}}{n^3} \left[ \frac{1}{E_0 - \frac{G_0}{n^2} - \hbar\omega - i\Gamma} + \frac{1}{2} \left\{ \frac{1}{E_0 + \Delta_0 - \frac{G_0}{n^2} - \hbar\omega - i\Gamma} \right\} \right], \quad (10)$$

where the parameters $A_{0x}$ and $G_0$ are the 3D exciton strength parameter and exciton binding [$G_0 = 34$ meV] energy, respectively.

### 3.3.2. $E_1$ and $E_1 + \Delta_1$ Transition

The $E_1$ and $E_1 + \Delta_1$ transitions in the zb-type semiconductors can take place along the [111] directions ($\Lambda$) or at L point in the BZ. These transitions are of the 3D $M_1$ type and occur in the zb InP at energies ~3.1 eV and ~3.31 eV, respectively. However, as the $M_1$ CP longitudinal effective mass is much larger than its transverse counterparts, one can treat these 3D $M_1$ CPs as a two-dimensional (2D) minimum $M_0$. The contribution to $\tilde{\varepsilon}(E)$ of the 2D minimum is given by [38]

$$\tilde{\varepsilon}^{b}(E) = -B_1\chi_1^{-1}\ln\left(1 - \chi_1^2\right) - B_2\chi_{1s}^{-2}\ln\left(1 - \chi_{1s}^2\right), \quad (11)$$

with

$$\chi_1 = (\hbar\omega + i\Gamma)/E_1, \quad (12a)$$

$$\chi_{1s} = \frac{\hbar\omega + i\Gamma}{(E_1 + \Delta_1)}, \quad (12b)$$

where B's and $\Gamma$ are the strength and broadening parameters, respectively, of energy transitions.

One must note that Equation (11) is a consequence of the one-electron approximation. Excitonic states should, in principle, exist at each type of CP, since the Coulomb-like interaction is always present between the electrons and holes. There may be only two analytical equations which enable us to treat the excitonic effects in the $E_1$ and $E_1 + \Delta_1$ spectral region: (i) the effective mass (EM) approximation and (ii) the Köster–Slater (KS) method [38]. Both the EM approximation and the KS method can dramatically modify and sharpen the $E_1$ and $E_1 + \Delta_1$ structures of semiconductors. We, however, find a better fit with experiments using the EM approximation rather than the KS method.

In the case of 3D $M_1$ CPs (i.e., the saddle-point excitons or hyperbolic excitons), the EM equation is much more difficult to solve. However, in the approximation of 2D $M_0$ CP, the equation gives a series of 2D Wannier-type excitons (discrete excitons). The contribution of these excitons to $\tilde{\varepsilon}(E)$ can now be written with Lorentzian line shape as [38]

$$\tilde{\varepsilon}^{b1}(E) = \sum_{n=1}^{\infty} \frac{1}{(2n-1)^3} \left[ \frac{B_{1x}^1}{E_1 - \frac{G_1}{(2n-2)^2} - \hbar\omega - i\Gamma_1} + \frac{B_{1sx}^1}{E_1 + \Delta_1 - \frac{G_1}{(2n-1)^2} - \hbar\omega - i\Gamma_2} \right], \quad (13)$$

where the $B_{1x}^1$ and $B_{1sx}^1$ are the 2D exciton strength parameters at the $E_1$ and $E_1 + \Delta_1$ saddle point excitons, respectively, and $G_1$ is the 2D exciton binding energy. The 2D EM approximation also gives a continuum part of the excitonic states. However, one can consider that the contribution of this part is like that of the one-electron approximation (i.e., Equation (10)).

### 3.3.3. $E_0'$, $E_0' + \Delta_0'$, $E_2$ and $E_1'$ Transitions

In the optical spectra of InP (InSb), the more pronounced structures found in the region higher than $E_1$ and $E_1 + \Delta_1$ energies [37,38] can be labeled $E_0'$, $E_0' + \Delta_0'$, $E_2$ and $E_1'$. The $E_0'$ and $E_0' + \Delta_0'$ energy transitions in the zb semiconductors are believed to take place either at the $\Gamma$ point or in the $\Delta$ direction near $\Gamma$ point (Figure 1). One must note that the nature of $E_2$

transition is more complicated as it does not correspond to a single, well-defined CP. The $E_1'$ transitions may take place, however, from the top of the valence band ($\Lambda_3^V$) to the second-lowest conduction band ($\Lambda_3^C$) near the L point. These CPs can be characterized following the idea of a damped harmonic oscillator (DHO) model and it is a good representation of the above transition energies [38]:

$$\widetilde{\varepsilon}^c(E) = \frac{C_2}{\left(1 - \chi_2^2\right) - i\chi_2\Gamma_2'},$$ (14)

$$\chi_2 = \frac{\hbar\omega}{E_2},$$ (15)

where $C_2$ represents the appropriate strength parameters and $\Gamma_2'$ represents the non-dimensional broadening parameters. Combining the contributions of different CPs to the total dielectric function $\widetilde{\varepsilon}(E)$ using Equations (8)–(14), one can write it as a sum of individual terms [38]:

$$\widetilde{\varepsilon}(E) = \widetilde{\varepsilon}^a(E) + \widetilde{\varepsilon}^{a1}(E) + \widetilde{\varepsilon}^b(E) + \widetilde{\varepsilon}^{b1}(E) + \widetilde{\varepsilon}^c(E) + \varepsilon_{1\infty},$$ (16)

where $\varepsilon_{1\infty}$ is the high-frequency (or the optical) dielectric constant. This term is assumed to be non-dispersive and may arise from other higher-lying gap contributions. The best-fitted MDF parameters reported in Table 2 are used for the calculations of optical constants for InP, InSb, and InP$_{1-x}$Sb$_x$ alloys (Sections 4.2.1 and 4.2.2).

Theoretical results reported in Sections 4.2.1 and 4.2.2 are assessed against the existing experimental (SE and PL) data [22,23].

## 4. Results and Discussion

We have reported our results of the systematic experimental and theoretical studies to comprehend the lattice dynamical, structural, and optical properties (Sections 4.1 and 4.2) of both the binary InP, InSb and ternary InP$_{1-x}$Sb$_x$ alloy [53–59] semiconductors. The outcomes of our findings on phonons and electronic characteristics are in reasonably good agreement with the experimental data. Some interesting features noticed in our simulations are compared/contrasted (Sections 4.1 and 4.2) in the following sections.

### 4.1. Raman Scattering

In Figure 2, we have reported our RT Raman scattering spectroscopy results recorded on GS-MBE-grown InP$_{1-x}$Sb$_x$/n-InAs samples (#T1, T2, T4, T5, see Table 1) by using a Reinshaw InVia spectrometer in the backscattering geometry in the 100–450 cm$^{-1}$ frequency range. The polarization-dependent Raman measurements (Section 4.1.1) of these samples are presented in Figure 3.

Obviously, the selected frequency region (100–450 cm$^{-1}$) in Figure 2 has covered the longitudinal optical $\omega_{LO}$ (transverse optical $\omega_{TO}$) modes of InSb ~193 cm$^{-1}$ ($\sim$ 185cm$^{-1}$) and InP phonons ~354 cm$^{-1}$ (~304 cm$^{-1}$), respectively. For zb materials, while the $\omega_{TO}$ modes are usually forbidden in the backscattering configuration of the (001) surface, the phonons can be observed as a weak feature either due to slight misalignment of the sample and/or alloy disorder caused by the breakdown of wavevector $\overrightarrow{\mathbf{q}}$ conservation rule [45]. The perusal of Figure 2 has clearly indicated InP-like $\omega_{TO}$ phonon as well as the defect-activated longitudinal acoustic (DALA) mode. In samples #T1, T2, while the DALA mode has occurred around ~147.2 cm$^{-1}$, it shifted to a lower value of ~143.4 cm$^{-1}$ for the #T4, T5 samples. An $A_{1g}$ phonon mode of antimony also appeared between ~160 and 165 cm$^{-1}$, as its intensity is much weaker compared to the DALA peaks. Two additional alloy-disordered related weak features are perceived in our Raman measurements near ~105 cm$^{-1}$ and ~129 cm$^{-1}$. Besides observing the InP-like $\omega_{LO}$ modes in #T1, T2, the Raman study has also detected InP-like $\omega_{TO}$ mode near ~298 cm$^{-1}$ which moved to a lower value at ~295 cm$^{-1}$ in #T4, T5 samples (Figure 2). In addition to the InP-like optical phonon

frequencies ($\omega_{LO}$, $\omega_{TO}$), we have also identified an additional disorder-activated optical (DAO) band appearing as a shoulder of the $\omega_{TO}$ mode between ~308 cm$^{-1}$ and 311 cm$^{-1}$.

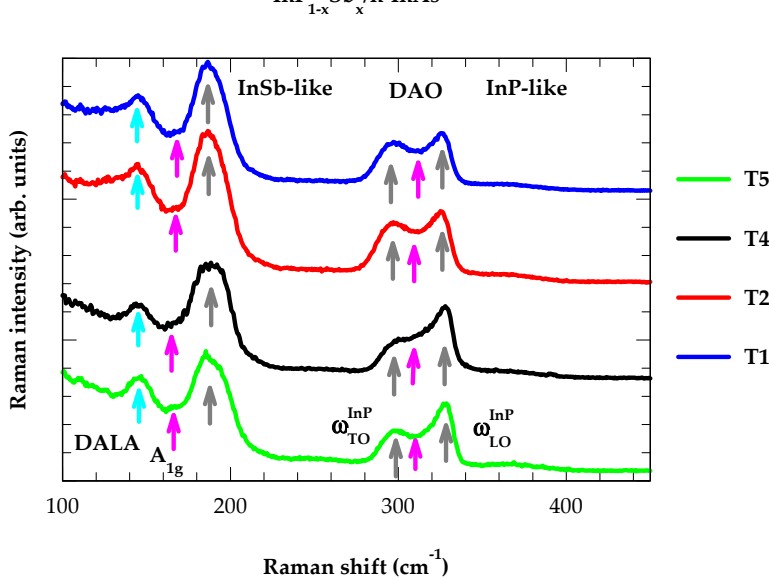

**Figure 2.** Results of our Raman scattering measurements in the backscattering geometry for four different GS-MBE-grown samples (see Table 1). Clearly, the Raman spectra have revealed only one set of composition-dependent InP-like $\left(\omega_{LO}^{InP}, \omega_{TO}^{InP}\right)$ modes. The InSb-like $\left(\omega_{LO}^{InSb}, \omega_{TO}^{InSb}\right)$ doublet has not been resolved. Disorder-activated longitudinal acoustic (DALA) and optical (DAO) phonons are also observed (see text).

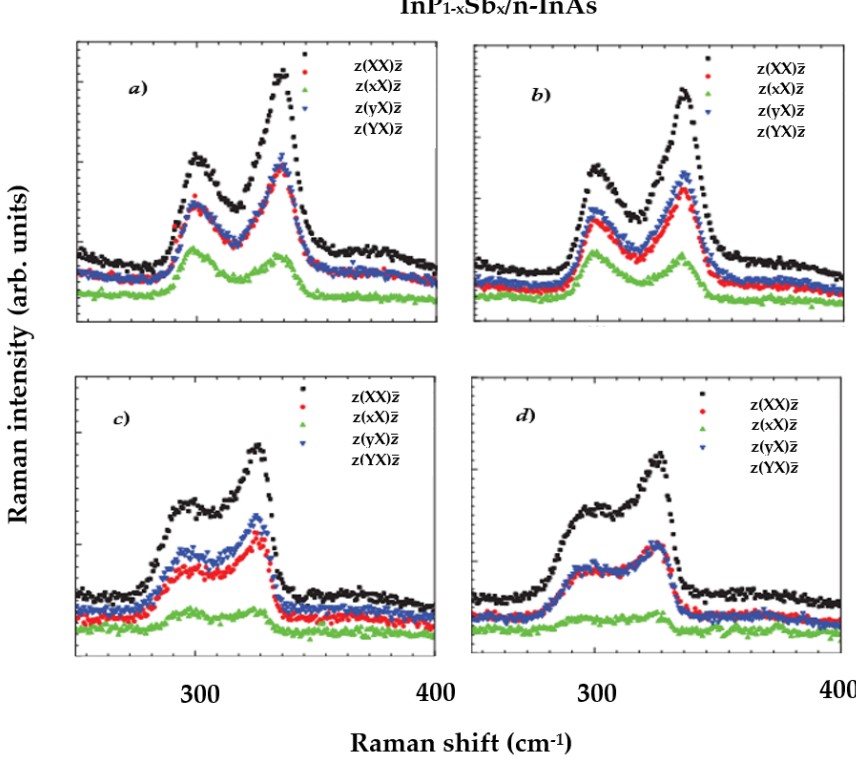

**Figure 3.** Polarization-dependent Raman scattering results of the composition-dependent InP-like $\left(\omega_{LO}^{InP}, \omega_{TO}^{InP}\right)$ optical modes for samples #T2 (**a**), T1 (**b**), T4 (**c**), and T5 (**d**) (see text).

Unlike observing the InP-like optical phonon doublets, we could only identify a single broad InSb-like band (Figure 2) near ~195 cm$^{-1}$ in GS-MBE-grown samples. Obviously, the InSb-like band in InP$_{1-x}$Sb$_x$ is not resolved, as its large (~10 cm$^{-1}$) full width at half maximum (FWHM) has a value larger than the separation of ($\omega_{LO} - \omega_{TO}$) ~8 cm$^{-1}$ in InSb, at x = 1 [49]. We strongly feel that the low-intensity InSb-like $\omega_{TO}$ band as well as the gap mode of In*P*:Sb near x → 0 are either buried in the vicinity of disordered optical bands and/or falling in the region where the density of phonon states is high [24–27]. In Section 4.1.2, we will address this issue theoretically by calculating the gap mode of Sb in In*P* and LVM of P in In*Sb* using a realistic lattice dynamical model in the ATM-GF framework [59].

### 4.1.1. Polarization-Dependent Raman Spectra

In Figure 3a–d, we have reported our results of polarization-dependent InP-like optical modes in the backscattering geometry for #T1, T2, T4, and T5 samples. Here, the polarization of the incident beam is fixed while analyzing the scattered light by using a Polaroid sheet combined with a half-wave plate, placing it in front of the entrance slit of laser light. The RSS spectra are recorded with two different polarization geometries z(XX)z̄ and z(YX)z̄, where X, Y, and Z denote the coordinates of (100), (010), and (001) directions with respect to the n-InAs substrate. In the polarization-dependent RSS, the characteristic of $\omega_{TO}$ in the backscattering geometry is a non-polar optical mode that carries intrinsic crystalline information. No matter how we rotate the InP$_{1-x}$Sb$_x$/n-InAs sample with respect to the z-axis, the intensity of $\omega_{TO}$ phonon remains constant [Figure 3a–d].

Like other Raman studies [24–27], our composition-dependent results have indicated that the InP$_{1-x}$Sb$_x$ ternary alloys exhibit a two-phonon-mode behavior (see Figure 4). We attribute this behavior to the strong bond bending in InPSb [40]. Again, in the limiting values of composition x for InP$_{1-x}$Sb$_x$, we are unable to identify the impurity vibrational modes (i.e., the gap mode of In*P*:Sb near x → 0 and LVM for In*Sb*:P near x → 1). While the LVM in P-doped InSb is observed by FTIR spectroscopy [43,44], no such measurement of the gap mode has been reported for In*P*:Sb. Our ATM-GF calculation (Section 4.1.2), providing accurate values of the impurity modes, has offered further justification for the two-phonon-mode behavior [53–59].

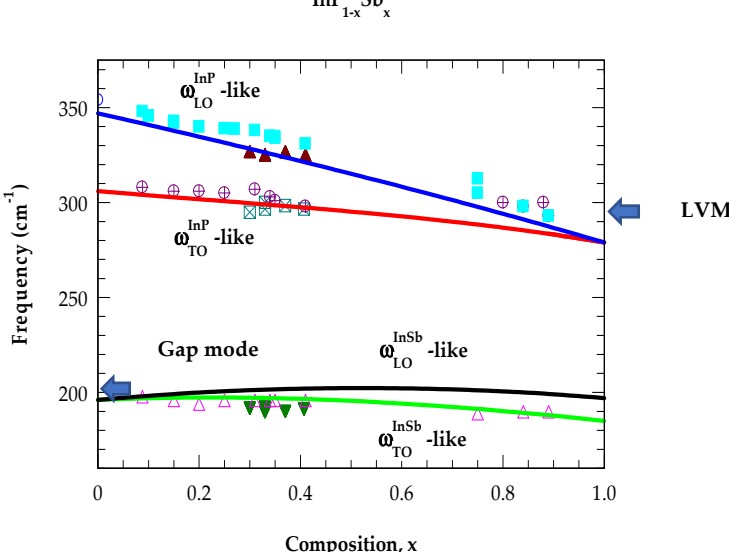

**Figure 4.** Comparison of the modified-random-element iso-displacement (MREI) model with our Raman scattering data for the two-phonon mode behavior of InP$_{1-x}$Sb$_x$ alloys. The blue arrows represent our results of LVM and gap mode using ATM-GF theory (see text).

4.1.2. Analysis of Optical Constants in the FIR → NIR Region

We have constructed the complex dielectric functions $\widetilde{\varepsilon}_L(\omega)$ for both the binary InP, InAs, InSb, and ternary $InP_{1-x}Sb_x$ alloys by using a classical "Drude-Lorentz" model. The optical constants are simulated by using Equations (3) and (4) in the frequency range of $400 \text{ cm}^{-1} \geq \omega \geq 100 \text{ cm}^{-1}$ by incorporating the phonons estimated from the RSS measurements (Sections 4.1 and 4.1.1). In Figure 5a–e, we have displayed $\omega$-dependent results of $R(\omega)$; $n(\omega), (\omega)$; and $\varepsilon_1(\omega), \varepsilon_2(\omega)$ for InP, InAs, InSb and $InP_{1-x}Sb_x$ (using x = 0.33, as an example).

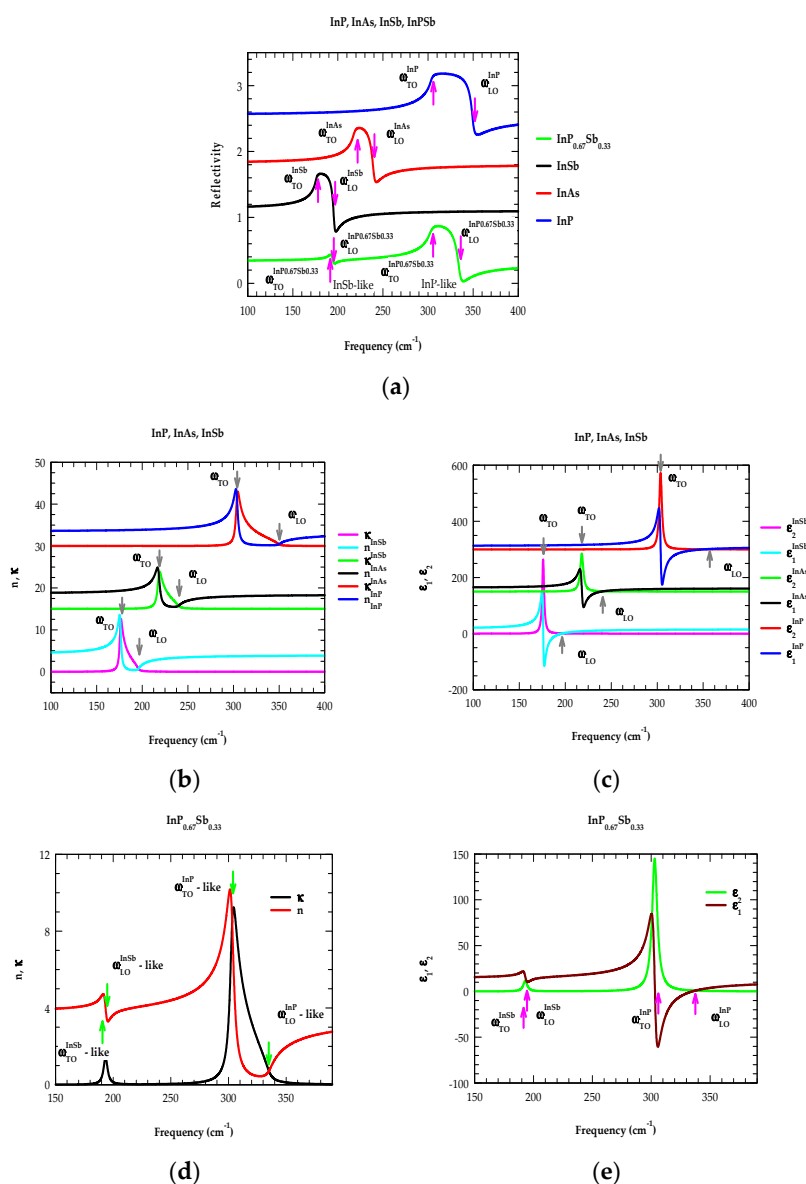

**Figure 5.** (**a**) Calculated frequency-dependent reflectivity spectra of phonons in the far-infrared region (between 100 and 400 cm$^{-1}$) for InP, InAs, InSb, and $InP_{0.67}Sb_{0.33}$ alloy based on Equation (5). The R($\omega$) results are shifted upward by 0.75 for clarity. The magenta vertical upward and downward arrows indicate transverse and longitudinal optical modes, respectively (see text); (**b**) same key as (**a**) but n($\omega$) and $\kappa$($\omega$) for binary InP, InAs, and InSb materials. The results are shifted by 15 for clarity; (**c**) same key as (**a**) but $\varepsilon_1$(E) and $\varepsilon_2$(E) for binary InP, InAs, and InSb materials. The results are shifted upwards by 150 for clarity; (**d**) same key as (**b**) but n($\omega$) and $\kappa$($\omega$) for ternary $InP_{0.67}Sb_{0.33}$ alloy; (**e**) same key as (**c**) but $\varepsilon_1$(E) and $\varepsilon_2$(E) for ternary $InP_{0.67}Sb_{0.33}$ alloy.

One must note that the polar materials reflect/absorb light in the FIR region due to the interaction of electromagnetic (EM) wave field with the $\omega_{TO}$ modes and coupling to the $\omega_{LO}$ phonons. For each binary material, the $\omega_{TO}$ mode can be seen as a peak in the $(\omega)$, $\varepsilon_2(\omega)$, and $R(\omega)$ spectra due to resonance interaction with EM wave field while the coupling of $\omega_{LO}$ modes appeared near $n(\omega) = (\omega)$, $\varepsilon_1(\omega) = 0$. For the ternary $InP_{0.67}Sb_{0.33}$ alloy, however, the perusal of Figure 5a,d,e have revealed results of the optical constants providing well-separated InP-like ($\omega_{LO}$, $\omega_{TO}$) modes while the InSb-like phonons fall very close to each other ($<3$ cm$^{-1}$).

For all GS-MBE-grown samples, the unresolved FWHM ~10 cm$^{-1}$ of InSb-like optical mode in $InP_{1-x}Sb_x$ is quite large. For x = 0.48, while the wide separation (see Figure 4) between InP-like ($\omega_{LO}$, $\omega_{TO}$) modes decreases, the split in InSb-like phonons increases but only slightly (~3–4 cm$^{-1}$ << FWHM). As stated before, the experimental results have not been able to identify the impurity vibrational modes (i.e., the gap mode for In*P*:Sb near x → 0 and LVM for In*Sb*:P near x → 1) in the limiting values of x. Following ATM-GF theory (described in detail elsewhere) [59] with appropriate perturbation models, we have calculated the LVM ~295 cm$^{-1}$ and a gap mode near ~205 cm$^{-1}$ (shown by blue color arrows in Figure 4), providing strong confirmation of the "two-phonon-mode" behavior.

*4.2. Analysis of Photoluminescence Spectra*

Reihlen et al. [26] performed low-temperature (10 K) PL and absorption measurements on several $InP_{1-x}Sb_x$ epilayers grown by OMVPE using InP, InAs, and InSb substrates. Large miscibility gaps were predicted [26] from x = 0.02 to 0.97 from the second derivative of Gibb's free energy [32]. We have estimated the energy bandgaps $E^{PL}$ at 13 K on GS-MBE-grown $InP_{1-x}Sb_x$/InAs samples having different compositions x by using a 532 nm DP-SSL excitation source. In Table 3, the results of our PL study are included (see Figure 6) for $0.10 \leq x \leq 0.48$ using solid black inverted triangles, green-filled circles, and light-blue-filled squares. Comparison is made with the absorption (open blue circles) and PL data (red open square) of Reihlen et al. [26]. Again, the x-dependent energy bandgaps (Figure 6, magenta solid line) of ternary alloys $E_g^{InPSb}(x)$ are calculated by adopting a second-order polynomial equation [18]:

$$E_g^{InPSb}(x) = E_g^{InP}(1-x) + E_g^{InSb}x - C_T x(1-x), \tag{17}$$

where $E_g^{InP}$, $E_g^{InSb}$ are the direct bandgaps of the bulk InP, InSb materials, and the term $C_T$ ($\equiv 1.9$) is used as a bowing parameter.

The perusal of Figure 6 has revealed two $E^{PL}$ energy phases (small red bracket) with values displayed by black inverted triangles, green solid spheres, and a nearly constant value of InAs (substrate) shown by sky-blue-colored squares. For $x \geq 0.30$, our PL study has provided only one $E^{PL}$ energy phase (large red bracket) indicated by black triangles. Earlier studies by Reihlen et al. [26] on $InP_{1-x}Sb_x$ epilayers have suggested large miscibility gaps between x = 0.02 and 0.97. Our results indicated that high-quality single-phase epilayers can be obtained around x ~ 0.3 in the center of the miscibility gap. The GS-MBE-grown epitaxial layer with composition x = 0.31 is lattice-matched to the InAs substrate. The strain energy caused by lattice mismatch has locally changed the range of miscibility, making the growth of a single crystal possible. Obviously, the OMVPE results cannot be fully applied to GS-MBE-prepared $InP_{1-x}Sb_x$/n-InAs samples, as the growth situation may differ from equilibrium conditions. Moreover, our MDF simulation for $InP_{0.67}Sb_{0.33}$ alloy provided the lowest energy gap $E_0$ ~ 0.46 eV, in good agreement with the existing first-principles calculation and the PL measurements.

**Table 3.** Values of the fitted parameters from the absorption spectra recorded by Reihlen et al. [26] (bandgap $E_g$ in eV) and photoluminescence peak energies ($E_{PL}$ in eV) at 10 K. We have also reported our PL results measured at 13 K on GS-MBE-grown InP$_{1-x}$Sb$_x$/n-InAs samples.

| Reihlen et al. (a) (10 K) | | | Our PL Data (b) (13 K) | | | | |
|---|---|---|---|---|---|---|---|
| x (a) | $E_g$ | $E_{PL}$ | x (b) | $E_{PL}^{II}$ | $E_{PL}^{II}$ | $E_{PL}^{I}$ | $E_{PL}^{InAs}$ |
| 0.034 | 1.309 | 1.172 | 0.1 | 0.981 | 0.741 | | 0.442 |
| 0.053 | 1.264 | 1.169 | 0.16 | 0.80 | 0.612 | | 0.444 |
| 0.067 | 1.225 | 1.129 | 0.17 | | | | |
| 0.086 | 1.17 | 1.059 | 0.30 | | | 0.485 | |
| 0.103 | 1.142 | 0.985 | 0.33 | | | 0.489 | |
| 0.122 | | 0.916 | 0.36 | | | | |
| 0.152 | 1.052 | 0.818 | 0.37 | | | | |
| 0.167 | 1.03 | 0.801 | 0.48 | | | 0.380 | |
| 0.177 | 1.002 | 0.752 | | | | | |
| 0.209 | 0.886 | 0.713 | | | | | |
| 0.257 | 0.834 | 0.594 | | | | | |
| 0.261 | 0.84 | 0.604 | | | | | |
| 0.299 | | 0.507 | | | | | |
| 0.31 | | 0.497 | | | | | |
| 0.369 | | 0.492 | | | | | |
| 0.42 | | 0.456 | | | | | |
| 0.423 | | 0.445 | | | | | |
| 0.95 | | 0.22 | | | | | |

(a) Ref. [26]. (b) Our PL data.

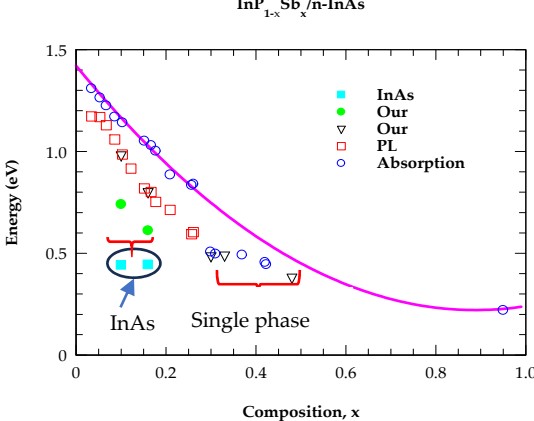

**Figure 6.** Comparison of x-dependent bandgap energies of InP$_{1-x}$Sb$_x$/n-InAs based on Table 3. Bandgap energies from absorption spectra (blue open circles) at 10 K (Ref. [26]) are fitted (magenta line) using Equation (16) with a bowing parameter of 1.9 eV. We have also displayed 10 K PL results open red squares and our 13 K PL data solid black triangles, green-filled circles, and light-blue-filled squares (see text).

### 4.2.1. Analysis of Optical Constants for Binary Materials in the NIR → UV Region

Optical constants, i.e., the real and imaginary parts of pseudo-dielectric functions $[\tilde{n}(E)\left(= \sqrt{\tilde{\varepsilon}(E)}\right)$ and $\tilde{\varepsilon}(E)]$, are numerically obtained by using the best-fitted MDF parameters (Table 2) following the methodology outlined in Section 3.3. For InSb, theoretical results of $n(E), \kappa(E)$ and $\varepsilon_1(E), \varepsilon_2(E)$ in the photon energy ($6\,\text{eV} \geq E \geq 0$) range are reported (red and blue lines) in Figures 7a and 7b, respectively, and compared with the SE data of Aspnes and Studna [22] and Seraphin and Bennett [23] (see black and green symbols). Similar results for InP are displayed in Figure 8a,b. The perusal of Figure 7a,b revealed that the sharpest peaks in $n(E)$ ($\varepsilon_1(E)$) are related to $E_1$ transitions while in $\kappa(E)$ ($\varepsilon_2(E)$), the strongest peaks are linked to $E_2$ transitions.

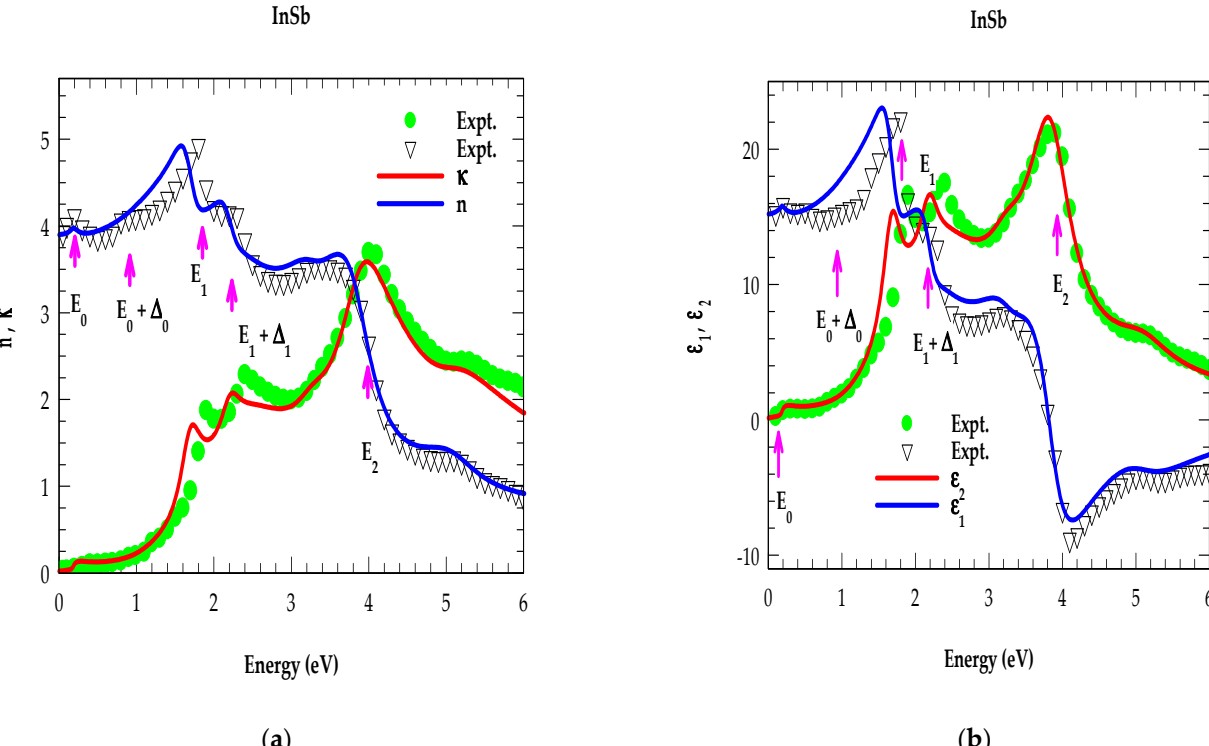

**Figure 7.** (**a**) Comparison of simulated MDF results (blue and red lines) for n(E) and κ(E) with experimental data (Refs. [22,23]) for InSb (indicated by symbols) in the energy range of 0–6 eV. Magenta vertical arrows indicate the critical point energies; (**b**) same key as (**a**) but for $\varepsilon_1(E)$ and $\varepsilon_2(E)$.

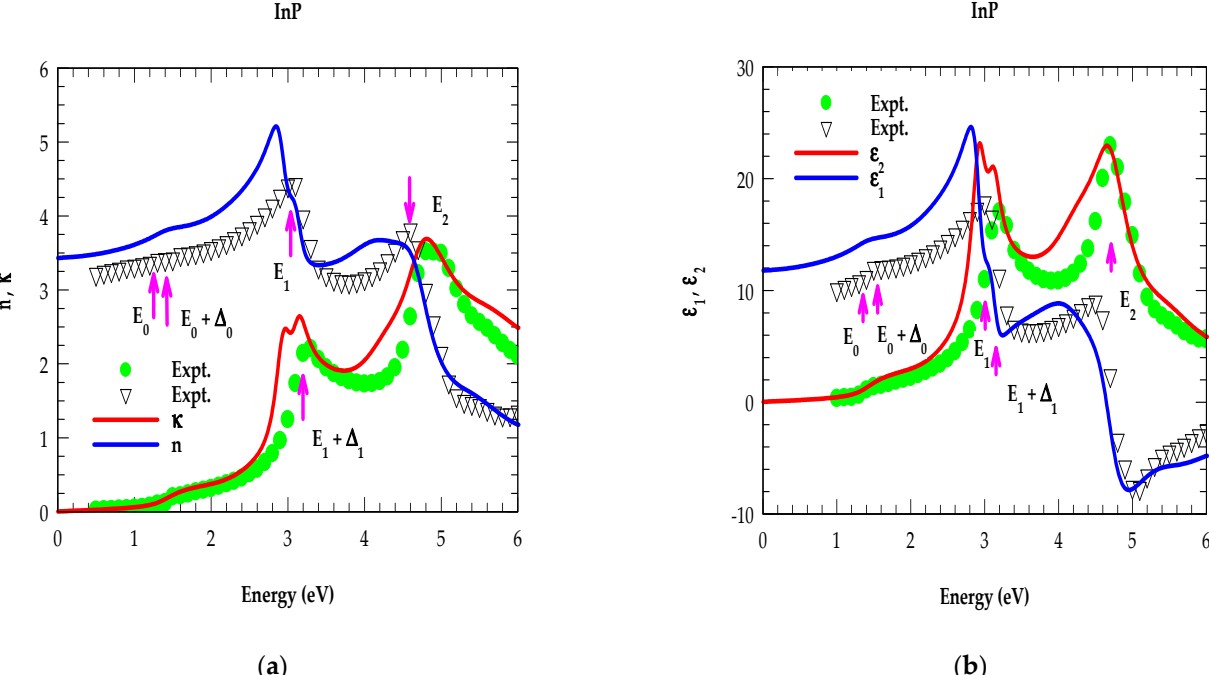

**Figure 8.** (**a**) Comparison of simulated MDF results (blue and red lines) for n(E) and κ(E) with experimental data (Refs. [22,23]) for InP (indicated by symbols) in the energy range of 0–6 eV. Magenta vertical arrows indicate the critical point energies; (**b**) same key as (**a**) but for $\varepsilon_1(E)$ and $\varepsilon_2(E)$.

For InSb and InP, our calculated results of reflectance R(E) and absorption $\alpha$(E) coefficients displayed in Figure 9a,b and Figure 10a,b have agreed reasonably well with the experimental results [22]. Interestingly, the simulations in each category has provided distinct CP energy features of the band structures arising from the inter-band transitions (indicated by magenta vertical arrows). Like n(E) and $\kappa$(E), the R(E) and $\alpha$(E) data have also revealed appropriate shifts in the major CP energy structures. Among other features, the reflectivity R(E) spectra indicated the strongest peaks at the $E_2$ energy gaps, varying between ~0.6 and 0.625 [see Figures 9a and 10a], in very good agreement with the observed values for the binary (InP, InSb) materials [22].

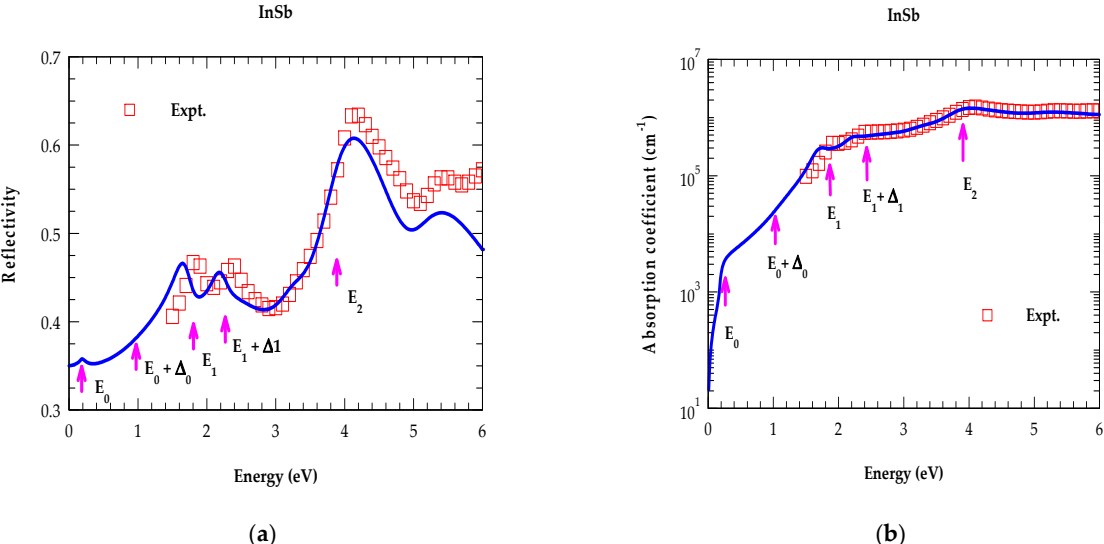

**Figure 9.** (**a**) Comparison of simulated MDF results (blue lines) for R(E) and $\alpha$(E) with experimental data (Ref. [22]) for InSb (indicated by red squares) in the energy range of 0–6 eV. Magenta vertical arrows indicate the critical point energies; (**b**) same key as (**a**) but for $\alpha$(E).

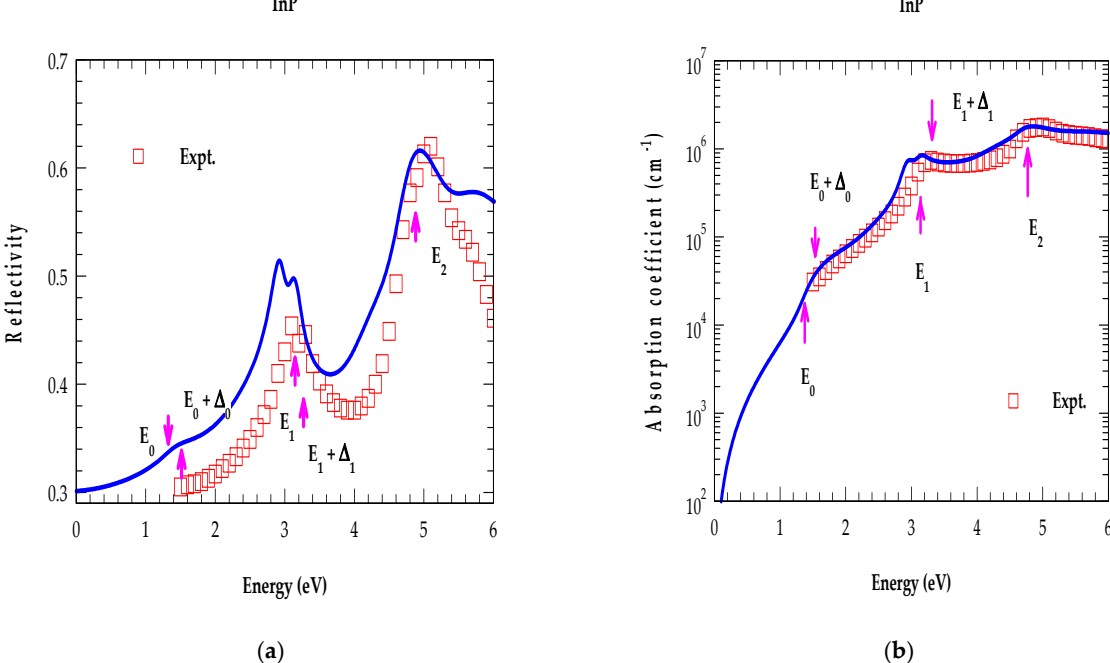

**Figure 10.** (**a**) Comparison of simulated MDF results (blue lines) for R(E) and $\alpha$(E) with experimental data (Ref. [22]) for InP (indicated by red squares) in the energy range of 0–6 eV. Magenta vertical arrows indicate the critical point energies; (**b**) same key as (**a**) but for $\alpha$(E).

4.2.2. Analysis of Optical Constants for InP$_{1-x}$Sb$_x$ Alloys in the NIR → UV Region

To calculate the composition-dependent optical constants for InP$_{1-x}$Sb$_x$ ternary alloys, we have meticulously obtained MDFs from the values (Table 2) of their binary counterparts (InP, InSb) by using appropriate quadratic expressions [18] involving x of the major CP energies, while the strength and broadening parameters are obtained deliberating a linear dependence articulation on x [38].

In Figure 11a–d, we have reported our simulated results of n(E), $\kappa$(E); $\varepsilon_1$(E), $\varepsilon_2$(E); R(E); and $\alpha$(E), respectively, for InP$_{1-x}$Sb$_x$ ternary alloys in the photon energy range of 6 eV ≥ E ≥ 0 by selecting the composition increment of x by 0.2. In each figure, the results have revealed an appropriate shift in the CP energies toward the low-energy side as x is increased, in corroboration with the observed changes from InP → InSb band structures. Like n(E), $\kappa$(E); $\varepsilon_1$(E), $\varepsilon_2$(E) (indicated by black and green points in Figure 11a,b the near normal-incidence reflectivity R(E) (shown by sky-blue-colored points) and the absorption coefficient $\alpha$(E) (represented by brown points) in Figure 11c,d also revealed distinct composition-dependent shifts in CP structures. Our simulations of optical constants for the InP$_{0.67}$Sb$_{0.33}$ alloy have provided the lowest energy bandgap value E$_0$ ~ 0.46 eV, in very good agreement with the existing first-principles calculation as well as our PL measurements (see Figure 6 and Table 3).

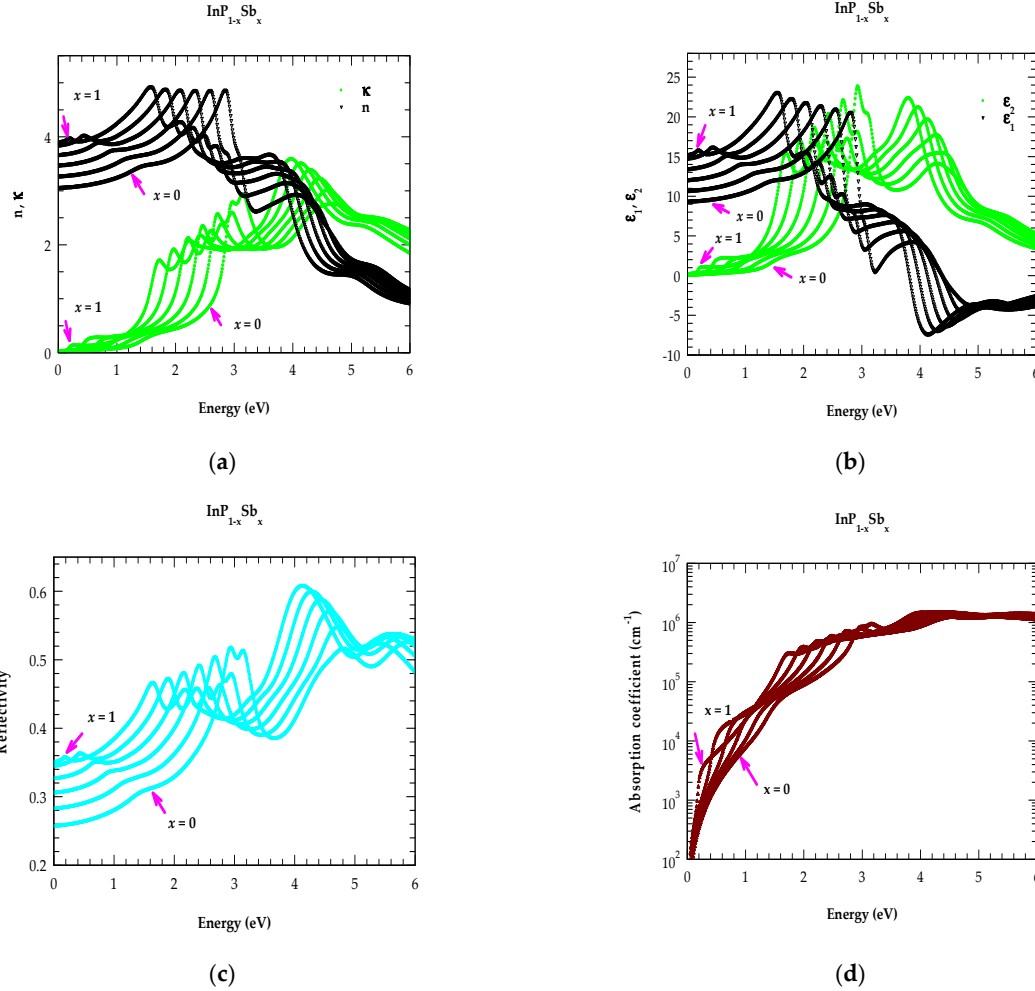

**Figure 11.** (**a**) Calculated n(E) and $\kappa$(E) spectra of InP$_{1-x}$Sb$_x$ alloys as a function of photon energy E with composition x, increment by 0.2. (**b**) Same key as (**a**) but for $\varepsilon_1$(E) and $\varepsilon_2$(E). (**c**) Same key as (**a**) but for reflectivity R(E). (**d**) Same key as (**a**) but for absorption coefficient $\alpha$(E).

## 5. Concluding Remarks

In conclusion, systematic assessments of phonon and optical characteristics are achieved both experimentally and theoretically for comprehending the vibrational, structural, and electronic behavior of high-quality GS-MBE-grown $InP_{1-x}Sb_x$/n-InAs (001) epilayers ($0.1 \leq x \leq 0.48$). To estimate the composition-dependent phonons, we have employed a Reinshaw InVia Raman spectrometer using a DPSS 532 nm laser as an excitation source. For bulk InP and InSb, the observed separations of ($\omega_{LO} - \omega_{TO}$) modes are 50 cm$^{-1}$ [48] and 8 cm$^{-1}$ [49], respectively. In each $InP_{1-x}Sb_x$/n-InAs (001) sample, our RSS measurements have identified InP-like $(\omega_{LO}^{InP}, \omega_{TO}^{InP})$ doublet and a DAO mode while revealing (Figure 2) InSb-like DALA, $A_{1g}$ Sb-impurity mode, and an unresolved broad optical band near ~195 cm$^{-1}$ with FWHM ~ 10 cm$^{-1}$. For x = 0.48, while the separation (see Figure 4) between InP-like ($\omega_{LO} - \omega_{TO}$) modes decreases, the split in InSb-like phonons increases, but only slightly. Our calculated separation between InSb-like ($\omega_{LO} - \omega_{TO}$) modes stays much smaller (~3–4 cm$^{-1}$ << FWHM) than the observed broad bandwidth—thus, resolving the $\omega_{TO}^{InSb}$ mode in $InP_{1-x}Sb_x$ is quite difficult [24–27]. Although the observed Raman line shapes of ternary $InP_{1-x}Sb_x$ alloys are signified as the optical phonon density of states exhibiting two-phonon-mode behavior, no LVM (In*Sb*:P; x → 1) and gap mode (In*P*:Sb; x → 0) have been perceived near the limiting values of x. Our ATM-GF calculations [59], providing accurate values of the impurity vibrational modes, have offered further support for the "two-phonon-mode behavior". Accurate results of bandgaps are achieved using PL measurements by calibrating the radiation lines of a Xe lamp and exploiting the SPEX 500M monochromator with a 532 nm DP-SSL excitation source. Although the earlier studies [26,27] on $InP_{1-x}Sb_x$ epilayers suggested large miscibility gaps from x = 0.02 to 0.97, our careful HR-XRD and PL results on $E^{PL}$ energy gaps have indicated attaining high-quality, single-phase epilayers around x ~ 0.3, ascribed to the effect of lattice match. A complete set of model dielectric functions is methodically achieved by using Adachi's MDF approach [37,38] for calculating the optical dispersion relations of both the binary InP, InSb and ternary $InP_{1-x}Sb_x$ alloys. For InP, InSb materials, the detailed analyses of optical constants (viz., n(E), $\kappa$(E); $\varepsilon_1$(E), $\varepsilon_2$(E); $\alpha$(E), and R(E)) over the photon energy range $0 \leq E \leq 6.0$ eV have exhibited reasonably good agreement with the SE experimental data [22,23]. The extension of MDF methodology to $InP_{0.67}Sb_{0.33}$ alloy has provided the lowest energy bandgap value $E_0$ ~ 0.46 eV, in very good agreement with the existing first-principles calculation [18] and PL measurements. We strongly feel that our systematic study, using RSS and PL measurements along with the MDF simulations of optical constants, has provided valuable information on the vibrational and optical characteristics of $InP_{1-x}Sb_x$/n-InAs (001) epilayers which can be extended to many other technologically important materials.

**Author Contributions:** D.N.T., Conceptualization, methodology, investigation, simulations, writing the original draft. H.-H.L., Supervision, GS-MBE growth of the $InAs_{1-x}N_x$/InP (001) samples, experimental data acquisition, editing. All authors have read and agreed to the published version of the manuscript.

**Funding:** This research received no external funding.

**Data Availability Statement:** The data that support the findings of this study are available from the corresponding author upon reasonable request.

**Acknowledgments:** D.N.T. wishes to thank Deanne Snavely, Dean College of Natural Science and Mathematics at Indiana University of Pennsylvania (IUP), for the travel support and the Innovation Grant that he received from the School of Graduate Studies making this research possible. H.-H.L. would like to acknowledge the financial support from the National Science Council, R.O.C, under Contract No. NSC 99-2221-E-002-105-MY3.

**Conflicts of Interest:** The authors declare that they have no financial interests/personal relationships that may be considered as potential competing interests.

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
