# Peer review of "Systematic Assessment of Phonon and Optical Characteristics for Gas-Source Molecular Beam Epitaxy-Grown InP1−xSbx/n-InAs Epifilms"

_crystals, doi:10.3390/cryst13091367_

Round 1

Reviewer 1 Report

Dear Editor,

I reviewed the manuscript entitled “Systematic assessment of phonon and optical characteristics for GS-MBE grown InP1-xSbx/n-InAs epifilms” by Devki N. Talwar and Hao-Hsiung Lin, finding it very interesting and worth to be published on  Thus Journal.

However, as a minor revision, I suggest the Authors to check the References section, since it seems tome that some papers (41, 47,48,49,50 and 51) are not cited in the full text.

Author Response

Authors response to Referee#1

  1. Thank you very much to the Crystals expert Referee for thoroughly reading our manuscript and for asking reasonable questions and/or for clarifications.

We must state that for all GS-MBE grown samples (including #T 7 not shown), the FWHM of InSb-like ~195 cm-1 optical mode stays as large as ~10 cm-1. By Raman spectroscopy the InSb-like modes are not resolved as the observed separation of wLO, wTO phonons at G point by INS [48] in InSb is ~8 cm-1 smaller than the FWHM. In InP1-xSbx alloys this separation at x = 0.48 between wLO, wTO modes become even smaller 3-4 cm-1 (see: Figure 4 “two-phonon mode behavior”). The disorder effects at higher x causes the large FWHM of InSb-like mode and makes it difficult to separate the wLO, wTO phonons. In this context appropriate changes are made in the revised manuscript by using red colored text and in the Concluding remarks.    

  1. The numbers and symbols in Figure 8 are too small and unclear, please enlarge them.                                                                      Thank you very much. We totally agree with your excellent suggestion. We have made these Figures 7 and 8 much larger – renumbered them as 7 – 8 and 9 – 10. The enhanced Figures would certainly help the reader to appreciate while comparing our simulated and experimental results on the optical constants. The large Figs. clearly indicate appropriate features of the critical point energies. Thanks for the suggestion.
  2. To make it clear to the readers, it is suggested that the author give detailed fitting results in Figure 7 and 8.                      Thank you very much. We have used red color text highlighting the detailed fitting procedures used in our simulations (see pages 6, 8) of the optical constants to obtain the results displayed in Figures 7-8 and 9-10. Thanks for the suggestion.
  3. Minor point: Line 161 “Polarization dependent Raman spectra has also been recorded (cf. Sec. 4.1.1)...”, pleases remove excess space. In addition, there are some repetitive descriptions in the article. For example, in section 3.1, “...[viz., refractive index n(w); extinction coefficient k(w); reflectivity R(w) and absorption coefficient α(w), etc.]...” and “ where n(w) is the refractive index and, k(w) an extinction coefficient or the attenuation index...”. Please recheck and revise the manuscript.

         Thank you very much. We have made appropriate changes, removed some repetitive words, and thoroughly checked all the References in the revised manuscript. Thanks for the suggestion

Reviewer 2 Report

In this manuscript, the author completed experimental and theoretical assessments of phonon and optical characteristics to understand the vibration, structure, and electronic behavior of GS-MBE grown InP1-xSbx/n-InAs samples. The theoretical data exhibited good agreement with experimental data. These results provide valuable information for the vibrational and optical traits of InP1-xSbx/n-InAs epilayers, and may provide a good reference for the growth of defect-free crystalline thin-films of ternary alloys. The overall quality of this manuscript is good and it can be accepted for publication in Crystals after minor revision.

1. The author mentioned in the article that ..., the InSb-like wLO, wTO modes are not resolved due to the large full width at half maximum (FWHM) of ~10cm-1 for ~195 cm-1 band..., but the author should include T7 sample as references. It is more convincing to observe the InSb-like wLO, wTO modes as the x increases.

2. The numbers and symbols in Figure 8 are too small and unclear, please enlarge them.

3. To make it clear to the readers, it is suggested that the author give detailed fitting results in Figure 7 and 8.

4. Minor point: Line 161   Polarization dependent Raman spectra has also been recorded (cf. Sec. 4.1.1)..., pleases remove excess space. In addition, there are some repetitive descriptions in the article. For example, in section 3.1, ...[viz., refractive index n(w); extinction coefficient k(w); reflectivity R(w) and absorption coefficient α(w), etc.]... and  where n(w) is the refractive index and, k(w) an extinction coefficient or the attenuation index.... Please recheck and revise the manuscript.

The author has a good level of English language proficiency, it is need that minor editing of English language required before publication

Author Response

Authors response to Referee #2

Thank you very much to your expert Referee#2 for thoroughly reading our manuscript. We are very happy to learn that you find our manuscript very interesting, and suitable for publication in this journal.  We have now carefully checked our manuscript and in the revised version made appropriate changes. Especially, we have checked that all the References cited in this manuscript are included at appropriate places.  All the changes are made by using red color text. 

Reviewer 3 Report

Intermetallic alloys and chemical compounds based on metals and non-metals of p-elements are extremely important from the point of view of science and technology. They exhibit a set of valuable applied properties related to optics, electrical conductivity, and magnetism. They are also valuable model environments for studying fundamental effects. Despite the large number of studies in this area, the variety of compositions determines the insufficient representation of works on this group of compositions in the open press. From this point of view, the presented study is timely and relevant.

The work is written in a neat, understandable scientific language. All data in the article are precise, summarized and discussed in relation to each other. The conclusions are based on the obtained results. The bibliography is representative, without inappropriate self-citations.

I have a small remark. Despite the fact that the work is very large, the introduction still occupies an inadmissible volume and is very difficult to read. There is no connection between some paragraphs. I would recommend significantly reducing it without reducing the number of literary references (literary edition).

In general, I unequivocally recommend this work for publication after the elimination of this small remark.

Author Response

Authors response to Referee #3

  1. Thank you to the Referee #3 of Crystals. However, the authors (DNT and HHL) strongly feel that the introduction part of the manuscript is important and relevant to the work that we have reported here in this manuscript. Some changes are made (using a red color text) in the manuscript which certainly help the reader to comprehend our work.
  2. The work is written in a neat, understandable scientific language. All data in the article are precise, summarized and discussed in relation to each other. The conclusions are based on the obtained results. The bibliography is representative, without inappropriate self-citations.                                                                            Thanks to the Referee #3. We highly appreciate the comment about the…. manuscript written in a neat, understandable scientific language. Thanks for the comments……on the data presented in the article to be precise, summarized and discussed in relation to each other. The conclusions are based on the obtained results. The bibliography is representative, without inappropriate self-citations.

3. In general, I unequivocally recommend this work for publication after the elimination of this small remark.

Thanks to the Referee #3 and for unequivocally recommending our work for publication.